# Development of Edible Composite Film from Fish Gelatin–Pectin Incorporated with Lemongrass Essential Oil and Its Application in Chicken Meat

**DOI:** 10.3390/polym15092075

**Published:** 2023-04-27

**Authors:** Farrah Azizah, Herwinda Nursakti, Andriati Ningrum

**Affiliations:** Department of Food and Agricultural Product Technology, Faculty of Agricultural Technology, Universitas Gadjah Mada, Flora Street no. 1, Bulaksumur, Yogyakarta 55281, Indonesia

**Keywords:** edible films, fish skin gelatin, composite films, lemongrass essential oil

## Abstract

One of the greatest challenges encountered by the food industry is the loss of quality of food products during storage, especially perishable foods such as chicken breast, which eventually adds to the waste. Edible films are known as a potential alternative to maintain food quality and also improve shelf life by delaying the microbial spoilage and providing moisture and gas barrier properties. Developments in edible films from biopolymer composites such as fish gelatin, pectin and essential oils have great potential and promising results in enhancing the shelf life of food products. This study was conducted to determine the effect of adding pectin and lemongrass essential oil on the properties of gelatin film and its application to preserve the quality of chicken breast. In this study, the fish skin gelatin and pectin were used with various compositions (100:0; 75:25; 50:50%), with and without the addition of lemongrass essential oil to develop edible films by a casting method. The results showed that the fish gelatin–pectin with the addition of essential oils caused a significant influence on several physicochemical properties such as the thickness, transmittance, transparency, water content, tensile strength, elongation at break and also antioxidant activity (*p* < 0.05). The antibacterial activity evaluation showed that edible film from a biocomposite of gelatin–pectin (75:25 and 50:50) with the addition of essential oil had an inhibitory effect on *Salmonella*. The biocomposite of the edible film made from gelatin–pectin and the addition of lemongrass essential oil have the potential to be developed as a food packaging material, especially for perishable food. Based on the result of the application of edible film to chicken breast, it also could maintain the quality of chicken breast during storage.

## 1. Introduction

Various types of natural polymers such as proteins and polysaccharides have been widely developed as environmentally friendly packaging both in the form of films and for food product coating applications, where the packaging has good properties as an oxygen carrier due to the hydrogen bond structure [1,2]. Several protein materials and derived materials have been developed into films, such as collagen, whey protein and gelatin. Gelatin is obtained from animal collagen through a hydrolysis process [3]. Generally, commercial gelatin is made from pork, but this is a problem for some people due to its socio-cultural and halal aspects. Therefore, nowadays, gelatin is produced from fish skin as an alternative [4,5,6]. The use of gelatin as a film material has weaknesses in terms of its resistance to water and sensitivity to moisture. To improve the properties and characteristics of the film, it is necessary to add other materials that can improve the properties of the film.

In addition to protein, the use of polysaccharides as a biodegradable film material has also been developed, one of which is pectin. Pectin has a structure of alpha 1–4 galacturonic acid and methyl ester galacturonic acid, which are commonly found in cell walls and fruit skins [7]. Pectin, which is applied as a film, is non-toxic and clear and has good mechanical strength and acts as a barrier for oil and oxygen, but on the other hand pectin does not have antimicrobial properties, and even becomes a carbon source for bacterial and fungal growth, as well as high water vapor permeability [8]. Mixing two or more polymers such as gelatin, pectin, starch or chitosan as film-making materials will create a material with new physical properties, and the manufacture of these composites is carried out to improve the properties of edible films with single components that have their respective weaknesses. For example, mixing pectin and gelatin can improve mechanical properties of the edible film [9]. Pectin–gelatin blended film offers advantages in terms of its mechanical properties, e.g., elongation at break, tensile strength and thermal properties concerning films formed from pectin or protein alone. The addition of orange peel pectin in fish gelatin films increased the hydrophobicity of the film, increased tensile strength and increased the antioxidant activity of edible film [10].

Due to the nature of the pectin material, which is a carbon source for microbes, the addition of antimicrobial compounds such as essential oils in edible films is a good alternative [11]. Gelatin and pectin have been selected for their low cost, widespread availability, non-toxicity and biodegradability. Among a multitude of biodegradable films investigated by our research laboratory, Gel/Pec-based film had the best mechanical and physicochemical properties [12].

Several essential oils can be incorporated into edible films to increase their active properties, one of which is lemongrass (*Cymbopogon citratus*) essential oil [13]. Lemongrass essential oil has good antioxidant and antimicrobial activity results and individually showed antibacterial activity on Gram-negative and Gram-positive organisms such as Escherichia coli and Staphylococcus aureus [13]. This study was conducted to determine the effect of adding pectin and lemongrass essential oil on the properties of gelatin film and its application to preserve the quality of chicken breast.

## 2. Materials and Methods

### 2.1. Material

The materials used in this study were fish skin gelatin (with a degree of bloom of 200 and 83 wt% protein) obtained from Redman (Singapore), pectin with a degree of esterification of 70% purchased from Ceamsa (Pontevedra, Spain), lemongrass essential oil (PT. Darjeeling Sembrani Aroma, Bandung, Indonesia), glycerol (Merck, Darmstadt, Germany) and Tween-20 (Merck, Darmstadt, Germany).

### 2.2. Edible Film Preparation

The formulations of various compositions of gelatin, pectin and lemongrass essential oil can be seen in Table 1. Powdered gelatin and powdered pectin were separately dissolved in distilled water with a concentration of 3% (*w*/*v*) at 50 °C for 30 min and stirred with a magnetic stirrer to form a good solution of film. Subsequently, 25% (*w*/*w*) of glycerol as plasticizer was added to the gelatin and/or pectin solution based on the polymer mass, mixing at 500 rpm for 5 min at 30 °C. For composite films, gelatin: pectin was mixed in various (*v*/*v*) ratios (100:0; 75:25; 50:50). Lemongrass essential oil (0; 0.5%) was prepared by mixing in 15% (*v*/*v*) Tween-20, the solution was stirred continuously using a magnetic stirrer at 500 rpm at 25 °C for 30 min and added to polymer (gelatin/pectin) solution. Then, homogenization was carried out with an Ultra Turax at 4000 rpm for 3 min and continued with the removal of the foam formed by allowing the solution to stand. After that, the film solution was poured into a pan with a size of 18 × 22 cm and a volume of 120 mL. Then, it was dried in an incubator at 45 °C for 30 h. The dry film was removed from the pan and conditioned at 25 ± 0.5 °C with a relative humidity of 50 ± 2% in a desiccator before testing.

### 2.3. Thickness

The thickness of the edible film with various concentrations of pectin–gelatin was tested using a micrometer screw that had been previously calibrated, with measurements made at 7 random points on the edible film.

### 2.4. Color

An objective color evaluation of the sample was performed using a Minolta Chroma (Model CR400, Osaka, Japan). The chromameter was set to the L*, a*, b* system and illuminant D65, with observer angle of 2° and aperture size 5.0 mm and a closed cone. Before the analysis, the chromameter was calibrated using a standardized white tile. The samples were measured in triplicate and averaged.

### 2.5. Water Content and Film Solubility

Water content and film solubility were measured according to [14], with modification. Films were cut to a size of 1 × 4 cm^2^ and put in a dry weighing bottle. The samples were dried at 105 °C for 24 h to reach a constant weight (*W*1). The sample was dissolved in 30 mL of distilled water and placed in a water bath shaker at 100 rpm for 24 h at room temperature. Furthermore, the sample was filtered with a Whatman filter. The filter containing the insoluble film was then dried in an oven at 105 °C for 24 h until a constant weight (*W*2).
(1)$$\mathrm{Film}\;\mathrm{Solubility}=\frac{\left(W1-W2\right)}{W1}\times 100$$
*W*1: initial weight. *W*2: weight after drying.

### 2.6. Transmittance and Transparency

Transmittance and transparency of film were measured according to previous methods, with modification. The film’s barrier properties to ultraviolet (UV) and visible light were measured by transmittance at the selected wavelength of 200–800 nm, using a UV–visible spectrophotometer. The film’s transparency value was calculated by:(2)Transparency=−logT600/x
T600 is transmittance at 600 nm and x is film thickness (mm).

### 2.7. Mechanical Properties (Tensile Strength and Elongation at Break)

Tensile strength and elongation at break analyses were performed using a universal testing machine (UTM). The film samples were cut to a size of 10 × 10 cm and a thickness of 0.5 cm at their midpoint. Films were placed on the UTM stretcher, then pulled by the device. The output of tests included F max data which were a measure of the tensile strength (MPa) and elongation at break (%). The tests were carried out three times on the middle and the edge of films [4].

### 2.8. Fourier Transform Infrared (FTIR) Spectra

Film samples in the form of powder were analyzed using FTIR spectra and recorded from wavenumbers 300–4000 cm^−1^ in the spectrometer. Spectra of the samples were obtained using a 0.50 mm thick potassium bromide disc. The discs were prepared by mixing 3–5 mg of powder films with 200 mg of dried potassium bromide. All spectra were acquired from 300–4000 cm^−1^. The obtained spectra were used to determine the interactions between components of the films. For each spectrum, 10 scans were added together [14].

### 2.9. Antioxidant Activities

The antioxidant activity was measured according to [15,16], with modification. The film was ground under liquid nitrogen. Then, 0.25 g of film powder was mixed with 5 mL of methanol (99%) and stirred for 5 min. It was centrifuged at 2700 rpm for 10 min at room temperature. Then, 1.5 mL film supernatant was added to 1.5 mL DPPH solution of 0.1 mM and incubated in the dark and at room temperature for 30 min. The absorbance was measured at a wavelength of 517 nm. Radical-scavenging activity (RSA) was calculated by the following Equation:(3)$$\mathrm{RSA}=\left(1-A_{\mathrm{Sample}}/A_{\mathrm{control}}\right)*100$$

### 2.10. Antibacterial Activities

The antibacterial activity was measured according to the previous method, with modification. The Escherichia coli and Staphylococcus aureus bacteria used had a concentration of 10^5^–10^6^ CFU/mL in 0.85% NaCl solution. First, 0.1 mL of bacteria solutions was spread on a nutrient agar (NA) plate. Film of 12 mm in diameter was placed on the surface of the NA plate and incubated at 37 °C for 24 h. The inhibition zone diameters were measured and the entire area of the clear zone was calculated [17].
(4)$$\mathrm{Inhibition}\;\mathrm{zone}=\frac{\left(D1-\mathrm{Ds}\right)+\left(D2-\mathrm{Ds}\right)}{2}$$
D1: vertical diameter. D2: horizontal diameter. Ds: film diameter.

### 2.11. Water Contact Angle Measurements

A water contact angle test to evaluate the hydrophobicity properties of edible film was carried out with an OCA20 Contact Angle analyzer. In brief, a 50 μL drop of distilled water was carefully deposited onto the surface of the film, and angles were measured in three different regions of each surface and averaged. In addition, an image of a distilled water drop was captured with a camera.

### 2.12. Microstructure Analysis of Films

Morphological characteristics of the surface and cross-section of the edible film were analyzed using scanning electron microscopy (JSM-6510LA, JEOL Ltd., Akishima, Tokyo).

### 2.13. Thermal Properties

The thermal properties of the gelatin–pectin composite film were analyzed using a differential scanning calorimeter (DSC-60Plus Shimadzu, Columbia, AR, USA). The melting point was measured in a constant stream of nitrogen at a flow rate of 10 mL/min. The analysis was carried out at a temperature range of −50 °C to 300 °C with a heating rate of 10 °C/min.

### 2.14. Application of Edible Film in Breast Meat Fillet

The edible film was used to wrap chicken breast meat fillets purchased at supermarkets in Yogyakarta. Chicken meat was cut into a size of 2 × 2 × 1 cm and divided into four groups, each group contained a portion of chicken meat packaged using composite film (G75M05, G50M05), commercial polyethylene plastic (PE) and without wrapping (control). Chicken meat was stored for 6 days at a temperature of ±5 °C in a cooling room. The sample was analyzed for changes in moisture content, color, PH and hardness at intervals of two days: days 0, 2, 4 and 6.

The moisture content of the chicken fillet was determined by drying the chicken meat in the oven at 105 °C for 24 h until a constant weight was obtained. Colors were measured using a CR-400 Chroma Meter (Konica Minolta, Japan). Every 2-day interval, a color measurement was carried out on the middle of the chicken meat. The color of chicken meat is expressed as the values of L* (light/bright), a* (reddish/greenish) and b* (yellowish/bluish). The hardness of chicken meat was measured by a universal testing machine (Zwick/Z0.5). The tested chicken sample was carefully placed in the center of the UTM instrument table. Then, the UTM needle dropped slowly until it was stuck into the chicken. All operations were automatically controlled by the UTM and the output of tests was obtained as F max data which were a measure of the hardness (N). All the hardness measurements were performed in three replicates.

The PH of chicken meat during storage was measured using a PH meter. A 5 g piece of chicken meat was homogenized in 50 mL of distilled water before measurement and afterward PH was determined.

## 3. Results and Discussion

### 3.1. Thickness

The thickness of the edible films was influenced by the different compositions of formulas such as the composition of gelatin, pectin and also the essential oil (Figure 1). The highest thickness is found in the formula with 50% gelatin, 50% pectin and 0.5% essential oil (G50M05). The lowest thickness of the edible film is found in the formula with 100% gelatin without any addition of pectin and essential oil. The thickness of the films depended on several factors, such as the hydrophobicity of EO incorporated in the edible films, surface tension and size of the oil droplet [16]. The thickness of the edible films in this study can be seen in Figure 1. This is in agreement with our previous research, in which the addition of clove essential oil and ginger essential oil increased the thickness of edible film [4].

### 3.2. Color

The color of edible films is shown in Table 2 which includes the values of L* (brightness), a* (redness) and b* (yellowness). Edible films without the addition of EO, especially the gelatin 100% formula, were brighter than those with the addition of EO and pectin. The brightness of edible films with lemongrass EO tends to be lower. The edible film a* value without EO was lower than with the addition of EO. The b* value fluctuated depending on the formula. The appearance of the edible film is shown in Figure 2.

### 3.3. Water Content and Solubility

The water content (WC) of tuna skin gelatin films is described in Table 3. The WC of edible film formulations is also influenced by pectin concentration and the addition of essential oil. Gelatin films with the addition of essential oil tend to have a higher WC, especially the formula of G100 and G100M05 (*p* < 0.05). On the other hand, the formulas of G75 and G75M05 and G50 and G50M05 were not significantly different in terms of WC (*p* > 0.05). Regarding solubility, the addition of EO tends to decrease the solubility of edible film in all formulas (*p* < 0.05). Film solubility in water is important for its potential use as food packaging. The solubility of edible films can be a parameter of the water resistance of the film. This is due to the interaction of the hydrophobic component of the lemongrass essential oil with the hydrophobic component of gelatin, thereby increasing the hydrophobicity of the edible film. This causes the solubility of edible films to decrease. A decrease in the solubility of edible gelatin films was also reported, as clove and ginger essential oils can reduce the solubility of the edible film [4]. Gel-based films incorporated with several EOs also showed similar results [12]. It was reported that this decrease could be attributed to the hydrophobic nature of the EO compounds and their interactions with hydroxyl groups of the film matrix.

### 3.4. Transmittance and Transparency

Transmittance and transparency are the optical properties of films that are quite important attributes that affect their appearance, market capabilities and suitability for various applications, such as on food surfaces. The transmittance of edible films from wavelengths of 200–800 nm has the pattern shown in Figure 3.

Transmittance at 350–800 nm wavelengths was relatively stagnant. Transmittance indicates the film’s barrier to light which can damage the product. The lowest transmittance was obtained from the addition of essential oils (EOs). Light transmission depended on the distribution of EO in the film’s matrix and the interactions between EO, pectin and gelatin that cause differences in the morphology of the film’s matrix with light transmission. EO can inhibit the transmission of light through films. This decrease in light transmission was probably caused by light scattering at the interface of EO droplets embedded in the film’s matrix [5].

The transparency of tuna skin edible films at a wavelength of 600 nm is shown in Table 4. Localized EO drops in the film matrix reduced the transparency of the gelatin film, most likely due to the effect of light scattering. Therefore, the incorporation of EO has an impact on the appearance and light barrier properties of the gelatinous film (Table 4). In transparent materials, non-uniformity in material composition causes significant changes in optical properties. Thus, the incorporation of EO in the films directly influences the light transmission and transparency of the resulting films.

### 3.5. Mechanical Properties (Tensile Strength and Elongation at Break)

The mechanical properties of biopolymers such as tensile strength and elongation at break depend on many factors such as the type and concentration of polymer matrix and additives (plasticizer, oil, cross-linking agent, filler), relative humidity and film forming conditions (type of solvent, drying rate) [11]. Tensile strength is the maximum pull that can be achieved until the edible film breaks. Tensile strength is a mechanical property of edible film. Tensile strength determines the strength of the edible film. The higher the tensile strength, the better the edible film can withstand mechanical damage [4]. The tensile strength (TS) test results are shown in Table 5. The TS of the edible films in this study ranges from 6.40–14.17 MPa. Table 5 shows that the addition of essential oil tends to increase the TS of edible films (*p* < 0.05). This indicates that EOs could act as cross-linking agents. Cross-linking was formed because molecules that have low molecular weights could more easily enter the gelatin tissue. Tuna skin gelatin was generally classified as having a high molecular weight (111.930–259.302 kDa) with a β (α chain dimer) component that showed cross-linking in the molecule [18]. Intermolecular interactions result in cross-links between chains, leading to improved film properties. This is in agreement with a study reported earlier where essential oils that contained mainly aldehyde, ketone and phenolic compounds, when interacting with protein films, increased the tensile strength of the film [4].

Elongation at break is the maximum length change experienced by the film until it breaks. The elongation at break (EAB) of the edible films in this study ranges from 24.96–86.13%, as shown in Table 5. The EAB of edible films was higher with the addition of essential oil (*p* < 0.05). That is because EO may act as a plasticizer and cause the films to become more plastic. EO containing monoterpenes of hydrocarbons will interfere with protein–protein interactions and reduce peptide bonds which are useful for stabilizing gelatin-based edible films, thereby increasing the stretchability of films. EO affects the strength of films. Previous research also showed that tilapia gelatin film which was incorporated with essential oils showed an increased EAB [19].

### 3.6. Fourier Transform Infrared Spectroscopy (FTIR)

In all edible film spectra (Figure 4), both gelatin films and pectin composites show absorption bands from 3100 to 3600 cm^−1,^ which are related to the vibration of hydrogen bonds from polysaccharides and proteins. Gelatin film showed major bands at approximately 3286 cm^−1^, 1630 cm^−1^, 1543 cm^−1^ and 1335 cm^−1^, corresponding to amide A (hydrogen bonding and N-H stretching), amide I (double bond C=O vibration of the COO- group bound to hydrogen), amide II (N-H bending vibrations and C-N stretching) and amide III (N-H vibration deformation and C-N stretching) [20]. In composite film, the addition of pectin to the gelatin film affects FTIR spectra. The addition of pectin material causes a shift in peaks, for example, in the regions of amide A, amide I and amide II where initially the peak of gelatin film (G100) was at 3286, 1630 and 1543 cm^−1^ and shifted to 3287, 1631 and 1544 in G75 (addition of pectin 25%) and to 3288, 1635, 1551 cm^−1^ in G50 (addition of pectin 25%). The peaks at 1630 and 1543 cm^−1^ that shifted with the addition of pectin are related to the gelatin α-helix, as its helical structure is affected by the addition of pectin [21]. One of the most visible is a peak at 1743 cm^−1^ which is only found in edible film with the addition of pectin, which indicates the stretching vibration of the carbonyl ester group (C=O) of the pectin material [21,22]. However, the peak at this wavenumber has a low intensity, because the pectin material has reacted with gelatin [23]. The FTIR spectral results of this composite film have confirmed that there is an interaction between pectin and gelatin.

### 3.7. Antioxidant Activities

Packaging films such as edible films should be able to effectively inhibit or slow down the rate of oxidation processes that cause food products to deteriorate and lose quality. Several emulsified essential oils gradually releasing antioxidants through natural active agents present in them are more preferred compared to the direct addition of synthetic antioxidants into food. Table 6 shows the DPPH radical-scavenging activity of the formulated films in this study. Lemongrass essential oil emulsion-based films act as a barrier between the food and the environment which significantly reduces the oxidation rate. Control films showed DPPH radical-scavenging activity although at much lower levels. This is because fish gelatin naturally has antioxidant activity due to several amino acids, such as glycine and proline, in the peptide bond that is formed during the film formation.

Polyphenolic compounds in essential oils are very important plant constituents, based on their antioxidant activity by binding to redox-active metal ions, deactivating lipid-free radical chains and preventing the conversion of hydro-peroxides to reactive oxy-radicals. Hydrogen atoms from the hydroxyl group (-OH) in phenols will donate electrons to bind to free radicals and prevent other components from oxidizing. The antioxidant activity of EO is not only linked to the presence of phenolic constituents; terpene alcohols, ketones, aldehydes, hydrocarbons and ether also contribute to the free radical-scavenging activity of some EOs. Several bioactive peptides in gelatin also have potential antioxidant activity that may contribute to the antioxidant activity of the film [6]. Increased antioxidant activity of gelatin films with added pectin occurs due to the presence of hydroxyl and uronil groups of monosaccharide units, carboxyl of the galactoriumnic acid group and the acetyl group released from the pectin chain during the extraction process, and the composite films can combine their bioactive compounds with antioxidant activity to cross-linked networks of pectin and gelatin [10].

### 3.8. Antibacterial Activities

Inhibition zones of edible films with EO for Salmonella and Staphylococcus aureus were larger than those without EO (Table 7). This was due to the presence of antibacterial compounds of lemongrass EO. The inhibition zone area could be classified into several groups related to antibacterial activity. Diameters >20 mm were classified as very strongly antibacterial, 11–20 cm was classified as strongly antibacterial, 6–10 mm was classified as moderately antibacterial and <5 mm was classified as weakly antibacterial [24]. Based on Table 7, the sample of edible film G75M05 showed moderate inhibition (9 mm inhibition zone) of *Salmonella*. On the other hand, the edible film sample G50M05 showed a very strong inhibition (14 mm inhibition zone). Figure 5 shows the inhibition zones of the samples.

When essential oils are added to edible films, essential oils are diffused into agar media and produce clear zones. Factors affecting the size of the inhibitory area include the sensitivity of the organism, culture medium, incubation conditions and the speed of agar diffusion.

The mechanism of the antimicrobial component contained in EO is a breakdown of the cytoplasmic membrane, which increases its permeability and depolarizes and causes cell leakage which can lead to bacteriocidal effects. EO is also able to cause damage to cell membranes and encourage cell lysis or help the release of cell wall autolytic enzymes that induce lysis [5].

### 3.9. Water Contact Angle

The water contact angle was determined to evaluate the hydrophilic/hydrophobic properties of the film without/with pectin and without/with lemongrass essential oil. Generally, a material is called hydrophobic if the contact angle formed between the surface and water is >90° (hydrophilic < 90°). In Figure 6, the angles formed on the film and water droplets are 50° to 80°, this shows the hydrophilic properties of edible film. The addition of pectin and lemongrass essential oil increases the contact angle to 80.18° in G75M05 (75% gelatin, 25% pectin and 0.5% lemongrass essential oil). This increase in the contact angle indicates an increase in the hydrophobicity properties of the edible film due to the interaction between gelatin and pectin through hydrogen bonds or electrostatic interactions. Previous results were similar, in which the addition of polysaccharide material (pectin and chitosan) increased the contact angle of gelatin film [10,25]. The addition of essential oils increases the value of the contact angle because the non-polar components of essential oils interact with hydrophobic gelatin compounds, and the presence of a dispersed hydrophobic phase, even at a small ratio, interferes with the hydrophilic phase and increases the tortuosity factor of mass transfer, so the hydrophobic properties of the film increase [26].

### 3.10. Microstructure of Films

Analysis by scanning electron microscopy (SEM) was carried out to determine the structure and microstructure changes of the surface and cross-section of the composite films. The results of the SEM analysis can be seen in Figure 7 (film surface) and Figure 8 (cross-section of film). Visible oil droplets can be seen on the surface of the film with the addition of lemongrass essential oil (Figure 8C) as well as cracking (Figure 8E,F). The same results were observed in previous research where the addition of Eucalyptus globulus essential oil and rosemary essential oil made the surface of the film rougher and oil droplets were seen on the film surface [7,27]. Cracking on the surface of the film occurred due to the incorporation of active compounds of a hydrophobic nature, causing the film to crack and lose structure, indicating that lemongrass essential oil was distributed in the film matrix with strong interactions [24,28].

In the SEM cross-sectional image, the film with the addition of lemongrass essential oil appears to have cavities/pores, like a sponge (Figure 8D–F). This could be due to the trapping of essential oil material in the polymer film which causes non-uniform aggregation and dispersion of lemongrass essential oil, making the film matrix be more open, with loose texture, and a lot of empty space and cracks [10,28].

The SEM results on the surface image of the edible film with the addition of pectin (Figure 7B,D) look similar to those of gelatin film (Figure 7A). This image indicates pectin and gelatin have high compatibility, resulting in a uniform film structure on the surface. However, on the edible surface of the film with composition of gelatin 75%–pectin 25% (Figure 7B), some parts look brighter, which is related to the content of phenolic compounds from pectin material, which transfers hydrophobic compounds to the surface of the film when drying, affecting the microstructure of edible film [24].

In the cross-sectional image, an edible film with gelatin content without pectin (Figure 8A,D) shows a compact structure compared to edible film with the addition of pectin (Figure 8B,C,E,F). The addition of pectin concentrations makes the structure of the complex film irregular, making the film coarse and wrinkled due to the difference in the physical properties of each polymer material that makes up the film [29]. The addition of low-methoxy pectin material to the gelatin film causes cracks in the film which can be attributed to the occurrence of phase separation caused by irregular gelatin coil–helix transitions and inhibits tissue formation due to excess polysaccharide molecules [30].

### 3.11. Thermal Properties

The results of the DSC analysis on edible films can be seen in Table 8 which shows the onset temperature (T0), peak temperature (Td) and enthalpy change (∆H). The T0 value of edible gelatin film was 130.98 °C, then it increased with the addition of pectin and lemongrass essential oil. After adding pectin and lemongrass essential oil, the T0 value of the film was 139.39 °C to 177.66 °C. When compared as a whole, the edible film with the addition of pectin and lemongrass essential oil had higher onset temperature, peak temperature (Td) and ∆H values, indicating higher thermal stability in the composite film.

Similar results have been reported before, that the addition of pectin increases the onset temperature, peak temperature (Td) and ∆H in pectin–casein–egg albumin composite film. The authors of other studies added pectin material to gelatin film to increase thermal stability. This occurs because of the molecular interactions between the gelatin and pectin polymers, so the films produced are more resistant to heat [2].

### 3.12. Application of Edible Film in Breast Meat Fillet

#### 3.12.1. Moisture Content

The moisture content of chicken meat decreased significantly (*p* < 0.05) during storage for all samples. The decrease in moisture content during the storage period occurs due to the process of evaporation of water from inside chicken meat to the environment to achieve water equilibrium due to the difference in vapor pressure in chicken meat and the environment [7]. Chicken meat coated with PE plastic can maintain its moisture content even until the sixth day due to the nature of PE plastic as a good water barrier. When compared to the controls, chicken meat with edible film loses significantly less water (*p* < 0.05) than uncoated chicken. Similar results were also reported where chicken meat coated with edible film had reduced water loss because the edible film layer on the surface of chicken meat functions as a semi-permeable barrier to reduce water transfer between chicken meat and the environment [31].

#### 3.12.2. PH Values

The PH value is one of the most important indicators for evaluating the quality and freshness of meat products, including chicken meat. The initial PH of chicken meat used in this study was 5.74 ± 0.09. The PH value increased significantly (*p* < 0.05) during the storage period for control meat and chicken meat with PE whereas the PH value of chicken meat with edible film only slightly changed from the initial PH. The increased PH value in the control and PE samples is due to the results of bacterial metabolism from protein degradation, the production of alkaline compounds in the form of ammonia (NH_4_) by microbes and growth of the bacterial population [32,33]. Meanwhile, samples wrapped with edible film (G75M05 and G50M05) were able to maintain the PH until the sixth day of storage. The edible film can prevent PH changes and protect chicken meat from microbial growth due to the antimicrobial activity of lemongrass essential oil and the presence of phenolic compounds in the composite polymer material [34,35].

#### 3.12.3. Hardness

The hardness of chicken meat increases significantly (*p* < 0.05) during the storage period for all samples except those with PE, while sample hardness on the second and fourth days did not change significantly (*p* > 0.05). The hardness value of chicken meat did not change with the edible films G75M05 and G50M05. When compared with all treatments, control chicken meat had the greatest hardness value (days 2, 4 and 6). The increase in hardness occurs due to water loss from chicken meat [31], according to the moisture content results. The ability of chicken meat with PE in maintaining moisture content and hardness is related to the nature of PE as a good water barrier. Nevertheless, a decrease in the value of hardness in chicken meat with PE on the fourth and sixth days of storage was accompanied by a change in the aroma of chicken meat and the water observed coming out of the surface of the chicken meat. The water on the surface of chicken meat with PE plastic can be a medium for the growth of bacteria and reduce the quality of chicken meat, followed by the change in the aroma of chicken meat during the observation process and the increase in PH value.

#### 3.12.4. Color Values

The color of the chicken meat on the first day was yellowish-white then began to become white and reddish. Meat color decreased significantly (*p* < 0.05) in brightness parameters (L*) for all samples (Table 9). The lower L* value indicates a darker color of the chicken meat. The resulting value of a* (redness) increased significantly (*p* < 0.05) during storage. On day 0 of storage, chicken meat had a negative reddish value, indicating the range of green color, and then at the end of storage the value became positive, indicating the presence of increased redness in chicken meat. The b* (yellowish) value increased significantly (*p* < 0.05) during storage for chicken meat with edible film packaging and decreased significantly (*p* < 0.05) for chicken meat with PE, while in chicken meat without packaging (control) the value of b* increased until the 4th day of storage and decreased on the 6th day of storage (Table 9). It should be noted that the edible film used was yellowish due to the color of pectin material and lemongrass essential oil.

## 4. Conclusions

Experimental results suggested that edible film, especially gelatin and pectin incorporated with lemongrass essential oil, can have functional properties including antioxidant activity and antibacterial activity that indicate its potential as an active packaging application for perishable foods such as chicken breast meat. The fish gelatin–pectin films with the addition of EO had significant differences in several physicochemical properties such as the thickness, transmittance, transparency, water content, tensile strength, elongation at break, antioxidant activity, water contact angle and also the microstructure. The antibacterial activity evaluation showed that edible biocomposite film made from gelatin–pectin (75:25 and 50:50) with the addition of EO had an inhibitory effect on Salmonella. The edible biocomposite film made from gelatin–pectin with the addition of lemongrass essential oil has the potential to be developed as a food packaging material, especially for perishable food such as chicken breast meat. Based on the result of the application of edible film to chicken breast, it could protect the quality of chicken breast during storage.

## Figures and Tables

**Figure 1 polymers-15-02075-f001:**
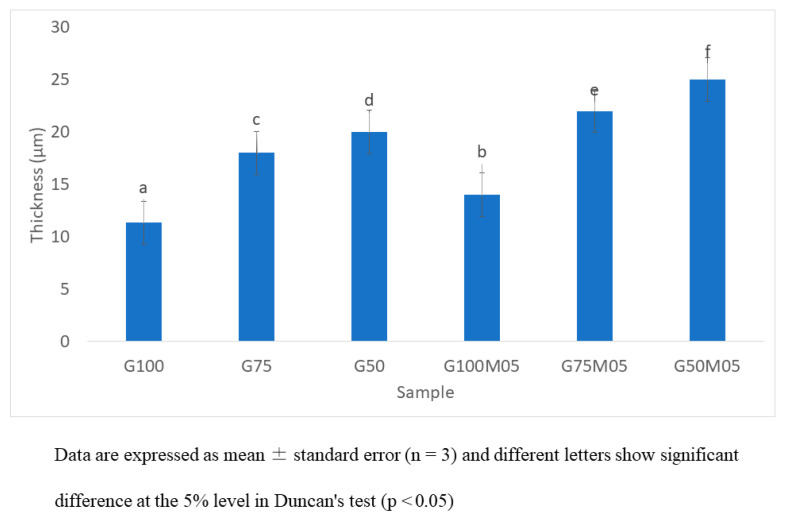
Thickness of edible film of fish skin gelatin–pectin and lemongrass essential oil.

**Figure 2 polymers-15-02075-f002:**
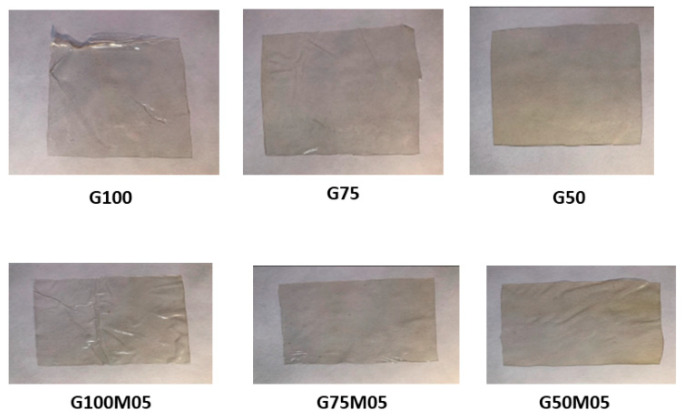
Appearance of Edible Film.

**Figure 3 polymers-15-02075-f003:**
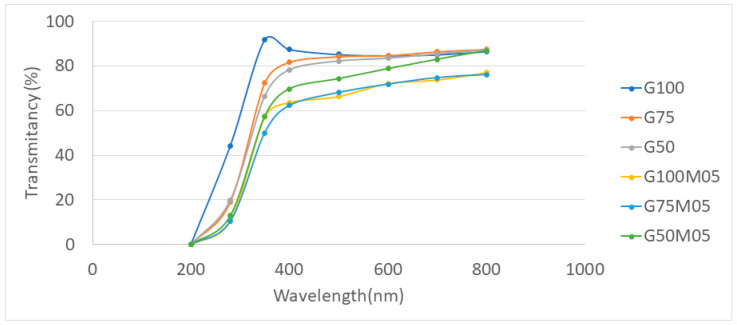
Transmittance of edible films.

**Figure 4 polymers-15-02075-f004:**
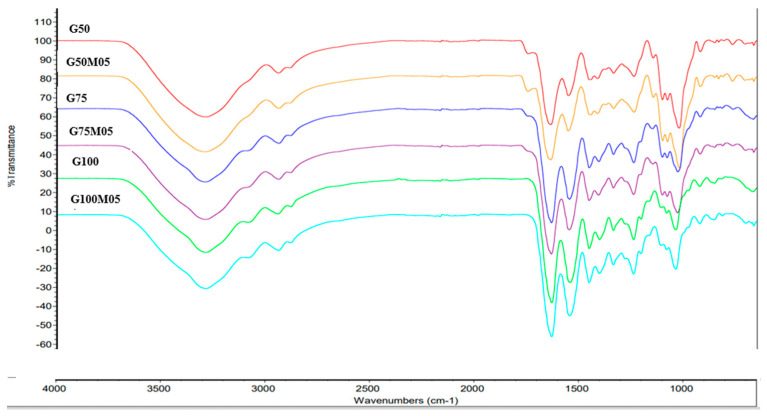
FTIR edible film of fish skin gelatin–pectin.

**Figure 5 polymers-15-02075-f005:**
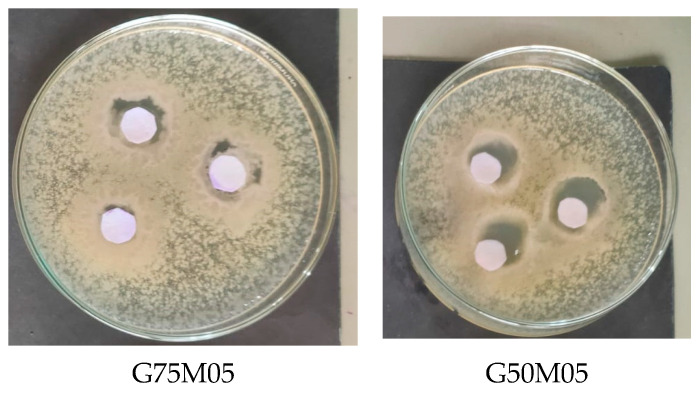
Antimicrobial activities of edible film of fish skin gelatin–pectin–essential oil.

**Figure 6 polymers-15-02075-f006:**
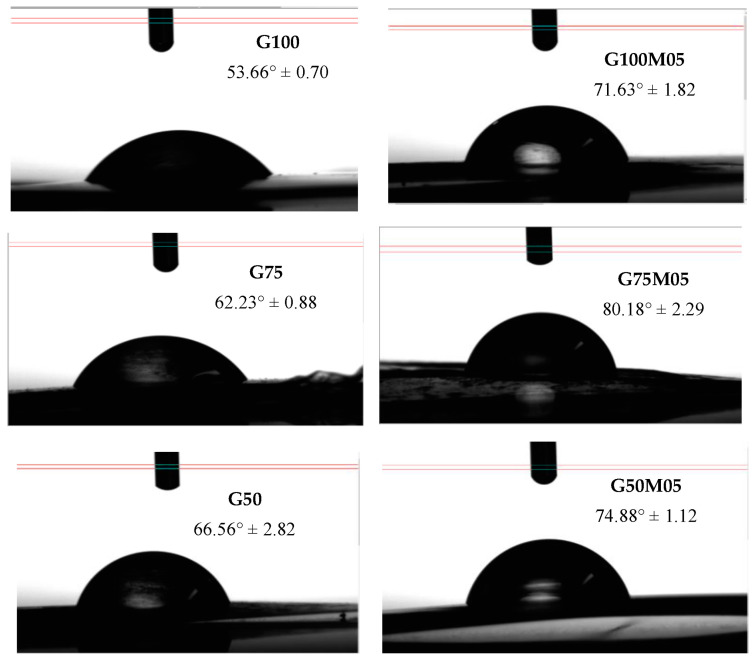
The water contact angle of edible film.

**Figure 7 polymers-15-02075-f007:**
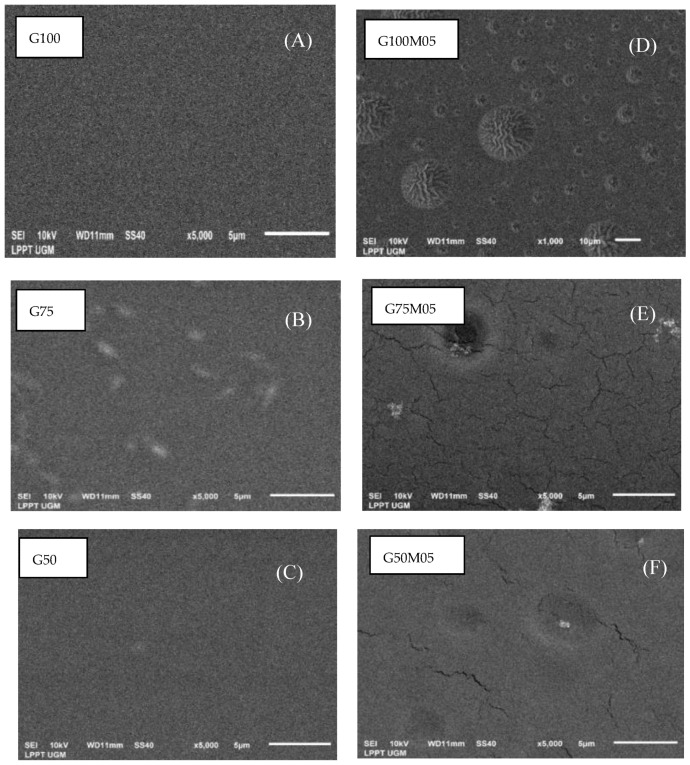
SEM micrographs of surface of edible films: (**A**) G100, (**B**) G75, (**C**) G50, (**D**) G100M05, (**E**) G75M05, (**F**) G50M05.

**Figure 8 polymers-15-02075-f008:**
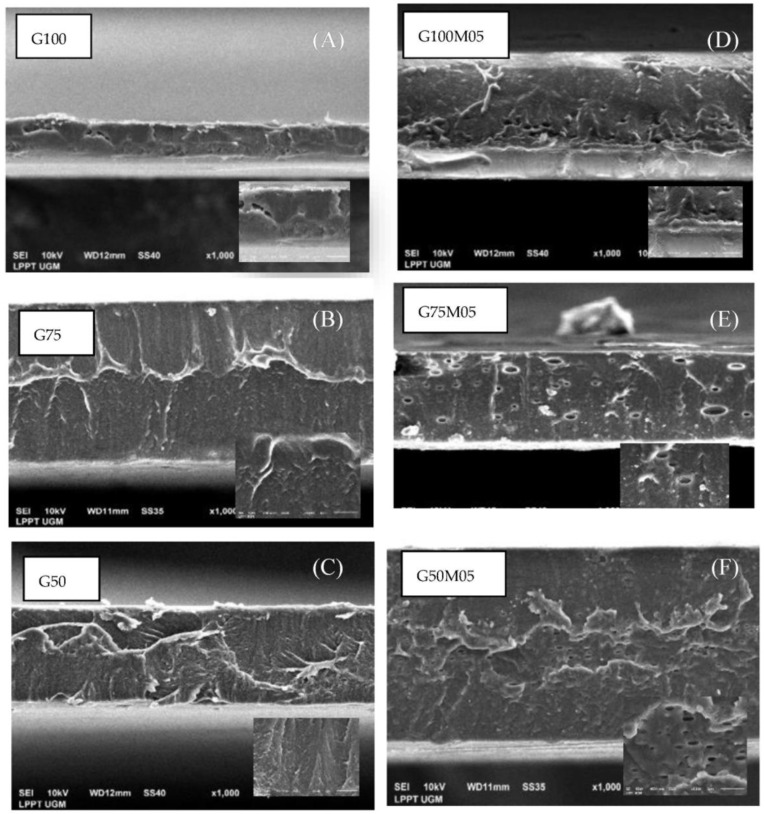
SEM micrographs of cross-sections of edible films: (**A**) G100, (**B**) G75, (**C**) G50, (**D**) G100M05, (**E**) G75M05, (**F**) G50M05.

**Table 1 polymers-15-02075-t001:** The formula of biocomposite film.

Formula of Biocomposite	Concentration of Lemongrass	Concentration of Glycerol	Sample Code
Gelatin 100% Pectin 0%	0%	25%	G100
	0.5%	25%	G100M05
Gelatin 75% Pectin 25%	0%	25%	G75
	0.50%	25%	G75M05
Gelatin 50% Pectin 50%	0%	25%	G50
	0.5%	25%	G50M05

**Table 2 polymers-15-02075-t002:** Color characteristics of fish skin–pectin edible film.

Sample	Color		
	L*	a*	b*
G100	93.11 d ± 0.1	−7.68 b ± 0.16	4.52 a ± 0.11
G75	91.72 bc ± 0.6	−7.89 a ± 0.03	5.33 b ± 0.20
G50	91.61 bc ± 0.15	−7.96 a ± 0.02	6.56 c ± 0.32
G100M05	91.97 c ± 0.57	−4.83 c ± 0.04	4.18 a ± 0.07
G75M05	91.21 ab ± 0.09	−4.55 d ± 0.14	5.14 b ± 0.31
G50M05	90.6 a ± 0.07	−4.03 e ± 0.07	5.24 b ± 0.57

Data are expressed as mean ± standard error (*n* = 3) and different letters in the same column show significant difference at the 5% level in Duncan’s test (*p* < 0.05).

**Table 3 polymers-15-02075-t003:** Water content and solubility in fish skin–pectin edible film.

Sample	Water Content (%)	Solubility (%)
G100	18.63 a ± 0.58	89.11 d ± 5.6
G75	21.78 c ± 0.18	69.18 c ± 2.8
G50	23.11 d ± 0.13	60.96 b ± 3.26
G100M05	19.57 b ± 0.36	71.16 c ± 1.15
G75M05	22.42 c ± 0.46	48.23 a ± 1.48
G50M05	23.57 d ± 0.28	56.87 b ± 1.45

Data are expressed as mean ± standard error (*n* = 3) and different letters in the same column show significant difference at the 5% level in Duncan’s test (*p* < 0.05).

**Table 4 polymers-15-02075-t004:** Transparency of edible film.

Sample	Transparency
G100	3.82 d ± 0.052
G75	3.69 c ± 0.101
G50	3.51 b ± 0.025
G100M05	3.60 b ± 0.017
G75M05	3.69 c ± 0.105
G50M05	3.39 a ± 0.023

Data are expressed as mean ± standard error (*n* = 3) and different letters in the same column show significant difference at the 5% level in Duncan’s test (*p* < 0.05).

**Table 5 polymers-15-02075-t005:** Tensile strength and elongation at break of edible film.

Sample	Tensile Strength (MPa)	Elongation at Break (%)
G100	9.55 a ± 3.93	75.01 cd ± 5.86
G75	6.37 a ± 3.99	54.75 b ± 17.46
G50	7.22 a ± 0.75	24.96 a ± 2.60
G100M05	14.17 a ± 1.49	86.13 d ± 1.75
G75M05	6.40 a ± 1.14	66.29 bc ± 16.37
G50M05	6.49 b ± 0.97	30.98 a ± 3.19

Data are expressed as mean ± standard error (*n* = 3) and different letters in the same column show significant difference at the 5% level in Duncan’s test (*p* < 0.05).

**Table 6 polymers-15-02075-t006:** Antioxidant activities of edible film.

Sample	Antioxidant Activity (%RSA)
G100	6.36 a ± 0.14
G75	11.84 b ± 0.14
G50	23.48 e ± 0.28
G100M05	13.25 c ± 0.20
G75M05	24.20 f ± 0.10
G50M05	20.36 d ± 0.20

Data are expressed as mean ± standard error (*n* = 3) and different letters in the same column show significant difference at the 5% level in Duncan’s test (*p* < 0.05).

**Table 7 polymers-15-02075-t007:** Antibacterial activities of edible film.

Sample	Inhibition Zone (mm)	
	*S. aureus*	*Salmonella*
G100	0	0
G75	0	0
G50	0	0
G100M05	0	0
G75M05	0	9
G50M05	0	14

Data are expressed as mean ± standard error (*n* = 3) and different letters in the same column show significant difference at the 5% level in Duncan’s test (*p* < 0.05).

**Table 8 polymers-15-02075-t008:** Thermal Properties.

Sample	Onset Temperature (°C)	Peak Temperature (°C)	∆H (J/g)
G100	130.98	131.65	−2.61
G75	139.39	142.94	−190.86
G50	162.78	163.72	−73.74
G100M05	170.96	171.71	−9.39
G75M05	177.66	179.06	−269.69
G50M05	173.88	176.64	−42.59

**Table 9 polymers-15-02075-t009:** Changes in moisture content, PH, hardness and color parameters of chicken breast meat samples at 0, 2, 4 and 6 days of storage.

Parameters	Days	Control	PE	G75M05	G50M05
Moisture content (%)	0	75.34 ± 0.21 ^dA^	75.37 ± 0.26 ^dD^	75.48 ± 0.24 ^dB^	75.09 ± 0.79 ^dC^
	2	60.85 ± 0.77 ^cA^	74.48 ± 0.07 ^cD^	62.65 ± 0.94 ^cB^	64.89 ± 0.66 ^cC^
	4	54.89 ± 0.73 ^bA^	74.29 ± 0.29 ^bD^	58.00 ± 0.74 ^bB^	61.6 ± 0.35 ^bC^
	6	44.12 ± 0.59 ^aA^	74.85 ± 0.11 ^aD^	46.38 ± 0.52 ^aB^	53.09 ± 0.3 ^aC^
PH	0	5.74 ± 0.09 ^bC^	5.75 ± 0.02 ^bD^	5.81 ± 0.02 ^bB^	5.8 ± 0.01 ^bA^
	2	5.63 ± 0.07 ^aC^	6.03 ± 0.07 ^aD^	5.68 ± 0.06 ^aB^	5.51 ± 0.03 ^aA^
	4	5.99 ± 0.15 ^cC^	6.44 ± 0.01 ^cD^	5.57 ± 0.02 ^cB^	5.43 ± 0.03 ^cA^
	6	6.45 ± 0.04 ^dC^	6.49 ± 0.03 ^dD^	5.67 ± 0.04 ^dB^	5.72 ± 0.07 ^dA^
Hardness (N)	0	0.63 ± 0.06 ^aC^	0.5 ± 0.03 ^aA^	0.46 ± 0.04 ^aB^	0.49 ± 0.07 ^aB^
	2	0.99 ± 0.05 ^bC^	0.61 ± 0.01 ^bA^	0.67 ± 0.05 ^bB^	0.62 ± 0.02 ^bB^
	4	1.15 ± 0.04 ^bC^	0.37 ± 0.04 ^bA^	0.73 ± 0.01 ^bB^	0.74 ± 0.08 ^bB^
	6	2.27 ± 0.11 ^cC^	0.36 ± 0.04 ^cA^	1.06 ± 0.04 ^cB^	1.07 ± 0.07 ^cB^
L*	0	53.82 ± 0.41 ^dA^	51.16 ± 0.86 ^dC^	54.52 ± 0.97 ^dA^	54.4 ± 0.69 ^dB^
	2	42.11 ± 1.59 ^cA^	49.51 ± 1.13 ^cC^	38.29 ± 1.39 ^cA^	42.97 ± 1.97 ^cB^
	4	37.66 ± 1.51 ^bA^	43.62 ± 1.38 ^bC^	38.98 ± 2.5 ^bA^	41.17 ± 2 ^bB^
	6	30.41 ± 0.12 ^aA^	47.94 ± 0.24 ^aC^	36.07 ± 0.2 ^aA^	36.55 ± 0.08 ^aB^
a*	0	−2.23 ± 0.11 ^aD^	−2.3 ± 0.09 ^aA^	−2.21 ± 0.1 ^aC^	−2.34 ± 0.18 ^aB^
	2	1.19 ± 0.04 ^bD^	1.32 ± 0.08 ^bA^	−1.97 ± 0.69 ^bC^	−1.94 ± 0.29 ^bB^
	4	2.35 ± 0.08 ^cD^	−1.27 ± 0.15 ^cA^	2.09 ± 0.55 ^cC^	1.19 ± 0.02 ^cB^
	6	3.02 ± 0.17 ^cD^	−1.57 ± 0.16 ^cA^	1.82 ± 0.69 ^cC^	1.79 ± 0.07 ^cB^
b*	0	11.22 ± 0.11 ^aB^	11.53 ± 0.71 ^aA^	11.66 ± 0.85 ^aC^	9.57 ± 1.47 ^aB^
	2	12.51 ± 0.42 ^bB^	9.29 ± 0.92 ^bA^	13.32 ± 0.7 ^bC^	11.85 ± 0.8 ^bB^
	4	13.57 ± 0.58 ^bB^	7.27 ± 0.51 ^bA^	14.36 ± 0.69 ^bC^	13.27 ± 0.81 ^bB^
	6	9.92 ± 0.03 ^aB^	7.57 ± 0.21 ^aA^	13.54 ± 0.06 ^aC^	13.29 ± 0.09 ^aB^

Data are expressed as mean ± standard error (*n* = 3) and different letters in the same column show significant difference at the 5% level in Duncan’s test (*p* < 0.05).

## Data Availability

Not applicable.

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
