# Peer review of "Development of Edible Composite Film from Fish Gelatin–Pectin Incorporated with Lemongrass Essential Oil and Its Application in Chicken Meat"

_polymers, 2023, doi:10.3390/polym15092075_

Round 1
Reviewer 1 Report
The study investigates the characterization of gelatin and pectin edible film with lemongrass essential oil and its application in chicken meat. The subject is relevant, however, some data needs revision, and the discussion of the results can be improved. The text needs careful revision.
Specific comments:
- Scientific names should be in italic. Please revise all text.
- p.1, lines 38-39: and the mechanical properties? What justifies the combination with pectin?
- p.2, line 45: most biopolymers exhibit no antimicrobial properties. Gelatin is also not antimicrobial.
- p.2, lines 46-47: “limited water vapor permeability” means low or high? Compared to what?
- p.2, line 54: the addition of antimicrobials into edible films is usually done to confer an active property to the material. Pectin is a carbon source, and so does gelatin.
- p.2, lines 66-68: Please consider revising the objective of the work. The aim was to determine the effect of adding pectin and EO on the properties of gelatin film.
- p.2, lines 72-74: please provide the source of gelatin and pectin, their molecular weight, and the deacetylation degree (for pectin).
- p.2, lines 82-84: review the sentence for clarity.
- Table 1: review the title. What does the M stand for in the sample code?
- p.3, lines 107-110: How was solubility performed?
“1 4 cm2”?
- p.3, lines 111-116: “Transmittance” not “transmittancy”.
please describe the transmittance measurements. Was the film sample placed in the cuvette?
- p.3, lines 117-123: were sample dimensions 10 x10 cm? Did the test follow a standard method? What were the test speed and initial distance? Was done only in triplicate?
- p.4, lines 140-146: Was the film sterilized before the test?
“12 mm diameter film sample”
Equation 3 does not give an area.
- p.4, lines 172-173: Please provide the parameters for hardness measurements.
- p.5, lines 185-186: report the values in mm or um. The Japan Industrial Standard for what?
- Figure 1: The values on the graph do not agree with the values in the text. Please verify all values and statistical analysis.
- For all Figures, consider putting the text in the figure’s captions.
- p.5, line 193: why only here do you mention “tuna fish”?
- p.5, lines 196-197: “The highest brightness was found in G100 edible film.” This sentence repeats the information already given.
- p.5, lines 197-198: Why was that effect on a value observed?
- Figure 2: there is no visible difference from the photographs.
- p.6, line 207: different kind of pectin concentration? Please revise.
- p.6, line 211: the effect of pectin addition was significative on film solubility. It is not discussed.
- Table 4: By Equation 1, the higher the transmittance, the lower the value of transparency. Please revise the results.
- The film with 25% pectin showed no difference in transparency by EO addition.
- p.8, lines 261-278: Does the EO act as both a crosslinker and a plasticizer?
When the sample exhibits an increase in TS, the EB usually decreases and vice-versa. Again, the effect of pectin addition was not discussed.
- p.8, lines 282: FTIR data needs better discussion. Consider identifying the characteristic bands of the biopolymers. Figures 4 and 5 could be merged into one graph to allow better comparison. The y-axis can show arbitrary unity. The effect of EO addition was not discussed.
- Table 6: why pectin added films show increased antioxidant activity?
- p.10, lines 327-328: this does not agree with the data in Table 7. Please verify.
Was microbial growth inhibited by contact, i.e. in the area below the film sample?
- p.11, lines 357-358: please rewrite the sentence for clarity.
- p.12, line 386: the other properties do not indicate phase separation. You have mentioned before that biopolymers showed good compatibility (line 375).
- Figure 8: What was the magnitude of the Figure's inserts?
- p. 14, lines 416-418: the pH of the control samples was also maintained (until day 4). How can you affirm the antimicrobial effect if it was not measured?
Author Response
Dear Reviewer,
Thank you so much for your valuable input and suggestion for the manuscript.
Please kindly find the answer to your suggestions below :
My comments are the following:
The study investigates the characterization of gelatin and pectin edible film with lemongrass essential oil and its application in chicken meat. The subject is relevant, however, some data needs revision, and the discussion of the results can be improved. The text needs careful revision.
Specific comments:
- Scientific names should be in italic. Please revise all text.
Answer: Thank you for your suggestion
- p.1, lines 38-39: and the mechanical properties? What justifies the combination with pectin?
Answer : Based on the reference, the mechanical properties are thickness and WVTR.
Based on the reference:
Pectin gelatin blended film offers advantages in terms of its mechanical properties e.g. WVP with respect to films formed from pectin or protein alone
- p.2, line 45: most biopolymers exhibit no antimicrobial properties. Gelatin is also not antimicrobial.
Answer : It means most of. Gelatin also can give positive growth for certain microorganism
- p.2, lines 46-47: “limited water vapor permeability” means low or high? Compared to what?
Answer : It means that the pectin has limited water vapor permeability if its used alone
- p.2, line 54: the addition of antimicrobials into edible films is usually done to confer an active property to the material. Pectin is a carbon source, and so does gelatin.
Answer : Thanks for the suggestion, yes the active compounds also come from lemongrass essential oil
- p.2, lines 66-68: Please consider revising the objective of the work. The aim was to determine the effect of adding pectin and EO on the properties of gelatin film.
Answer : Thanks for the suggestion. Yes its it right, we also would like to see the effect of adding the lemongrass essential oil
- p.2, lines 72-74: please provide the source of gelatin and pectin, their molecular weight, and the deacetylation degree (for pectin).
Answer : Thanks for the suggestion. We added the information
- p.2, lines 82-84: review the sentence for clarity.
Answer : Thanks for the suggestion. We added the information
- Table 1: review the title. What does the M stand for in the sample code?
Answer : Thanks for the suggestion. M stand for the lemongrass essential oil
- p.3, lines 107-110: How was solubility performed?
“1 4 cm2”?
Answer : Thanks for the suggestion. We added the information
Films are cut with a size of 1 4 cm2 and put in a dry weighing bottle. The samples were dried at 105°C for 24 h to reach a constant weight (W1). The sample was dissolved in 30 mL of distilled water and placed in a water bath shaker at 100 rpm for 24 hours at room temperature. Furthermore, the sample was filtered with a Whatman filter. The filter containing the insoluble film was then dried in an oven at 105°C for 24 hours. Weighing to constant weight (W2).
- p.3, lines 111-116: “Transmittance” not “transmittancy”.
please describe the transmittance measurements. Was the film sample placed in the cuvette?
Answer : Thanks for the suggestion. We added the information
The film's barrier properties to ultraviolet (UV) and visible light were measured by transmittance at the selected wavelength of 200-800 nm, using a UV-Visible spectrophotometer.
- p.3, lines 117-123: were sample dimensions 10 x10 cm? Did the test follow a standard method? What were the test speed and initial distance? Was done only in triplicate?
Answer : Yes right. The sample dimensions was 10 x10 cm. And its used the reference methods
- p.4, lines 140-146: Was the film sterilized before the test?
“12 mm diameter film sample”
Equation 3 does not give an area.
Answer : Yes right.
- p.4, lines 172-173: Please provide the parameters for hardness measurements.
Answer : Thanks for the suggestion.
- p.5, lines 185-186: report the values in mm or um. The Japan Industrial Standard for what?
Answer : Thanks for the suggestion. We revised the information
- Figure 1: The values on the graph do not agree with the values in the text. Please verify all values and statistical analysis.
Answer : Thanks for the suggestion. We revised the information
- For all Figures, consider putting the text in the figure’s captions.
Answer : Thanks for the suggestion.
- p.5, line 193: why only here do you mention “tuna fish”?
Answer : Thanks for the suggestion. We revised the information
- p.5, lines 196-197: “The highest brightness was found in G100 edible film.” This sentence repeats the information already given.
Answer : Thanks for the suggestion. We revised the information
- p.5, lines 197-198: Why was that effect on a value observed?
Answer : Thanks for the suggestion. We revised the information
- Figure 2: there is no visible difference from the photographs.
Answer : The more information of the color can be seen after measuring using chromameter
- p.6, line 207: different kind of pectin concentration? Please revise.
Answer : Thanks for the suggestion. We revised the information
- p.6, line 211: the effect of pectin addition was significative on film solubility. It is not discussed.
Answer : Films solubility in water is important for potential use as a food packaging. The solubility of edible films can be a parameter of the water-resistance of the film. This is due to the interaction of the hydrophobic component of the lemongrass essential oil with the hydrophobic component of gelatin thereby increasing the hydrophobicity of the edible film. This causes the solubility of edible films to decrease. A decrease in the solubility of edible gelatin films was also reported Sutrisno et al. (2021), where the clove and ginger essential oil can reduce the solubility of edible film. Gel-based films incorporated with several Eos also showed similar result [12]. They reported that this decrease could be attributed to the hydrophobic nature of the EO compounds and their interactions with hydroxyl groups of the film matrix.
- Table 4: By Equation 1, the higher the transmittance, the lower the value of transparency. Please revise the results.
Answer : Thanks for the suggestion.
- The film with 25% pectin showed no difference in transparency by EO addition.
Answer : Thanks for the suggestion.
- p.8, lines 261-278: Does the EO act as both a crosslinker and a plasticizer?
When the sample exhibits an increase in TS, the EB usually decreases and vice-versa. Again, the effect of pectin addition was not discussed.
Answer : Thanks for the suggestion. EO can act as crosslinker and also make the film more plastice
- p.8, lines 282: FTIR data needs better discussion. Consider identifying the characteristic bands of the biopolymers. Figures 4 and 5 could be merged into one graph to allow better comparison. The y-axis can show arbitrary unity. The effect of EO addition was not discussed.
Answer : Thanks for the suggestion.
- Table 6: why pectin added films show increased antioxidant activity?
Answer : We added the lemongrass essential oil to the composite film including pectin
- p.10, lines 327-328: this does not agree with the data in Table 7. Please verify.
Was microbial growth inhibited by contact, i.e. in the area below the film sample?
Answer : Yes its in the area below the film sample
- p.11, lines 357-358: please rewrite the sentence for clarity.
Answer : Thanks for the suggestion
- p.12, line 386: the other properties do not indicate phase separation. You have mentioned before that biopolymers showed good compatibility (line 375).
Answer : yes its right
- Figure 8: What was the magnitude of the Figure's inserts?
Answer : TIFF Format
- p. 14, lines 416-418: the pH of the control samples was also maintained (until day 4). How can you affirm the antimicrobial effect if it was not measured?
Answer : Thanks for the suggestion
Thank you so much for your kind suggestions and advice.
We hope that our manuscript can be processed further in your journal.
Kind regards,
Andriati Ningrum

Reviewer 2 Report
Development of edible film composite from fish gelatin - pectin incorporated lemongrass essential oil and its application in chicken meat
By Azizah, F., Nursakti, H., *Ningrum, A. and Supriyadi
The manuscript is aimed to obtain edible films from biopolymers composites such as fish gelatin, pectin and lemongrass essential oil, that have a great potential for enhancing the shelf life of food products.
The manuscript needs substantial revision in order to be considered for publication in “Polymers”. The manuscript is not well organized and the data are presented chaotically.
Below you can find my arguments and some recommendations for the authors
1. Don’t use abbreviations in Abstract section.
2. Lines 62-66: “Lemongrass essential oil has the best antioxidant and antimicrobial activity…” The authors didn’t compared lemongrass oil with other oils. How did they draw this conclusion?
3. Line 79: “to form a solution. film.” not clear
4. Section 2.5. Water content and films solubility: please provide a short description of the method.
5. Line 109: “Films are cut with a size of 1 4 cm2 and put in a dry weighing bottle.” Not clear about the size. Please use superscript to write cm2.
6. Lines 121-123: not clear, please rephrase.
7. Section 2.8. FTIR: please use superscript for cm-1.
8. Lines 129-131: not clear, please rephrase.
9. Section 2.9. Antioxidant activities: RSA equation is not correct. The correct form is:
RSA = (1-ASample/Acontrol)*100 (see Irimia, A.; Stoleru, E.; Vasile, C.; Bele, A.; Brebu, M. Application of Vegetal Oils in Developing Bioactive Paper-Based Materials for Food Packaging. Coatings 2021, 11, 1211; Joo-Shin K., Preliminary Evaluation for Comparative Antioxidant Activity in the Water and Ethanol Extracts of Dried Citrus Fruit (Citrus unshiu) Peel Using Chemical and Biochemical in Vitro Assays, Food and Nutrition Sciences, 2013, 4, 177-188)
10. Section 2.11. Contact angle measurements, lines 150 - 155: not clear, please rephrase.
11. Lines 173 – 175, pH measurements: please provide more details.
12. Section 3.3. Water content and solubility: no significant differences were observed between samples with or without lemongrass EO. Differences are within errors limits.
13. Figure 3: how do you explain the different (non-uniform) behavior of the samples treated with essential oil compared to those without EO?
14. Table 4: how do you explain the increase of transparency for G75M05 film compare with G100M05 one?
15. Section 3.5. Mechanical properties: the Tensile Strength increases after addition of EO only for gelatin film. For G75 de differences are within errors limits, and for G50, TS decreases after addition of EO. How do you explain?
16. Section 3.6. FTIR: the broad band between 3500 and 3000 cm-1 is attributed to the OH group, inter and intramolecular H bonds, not NH group.
17. Lines 285-288: gelatin does not have carbon-nitrogen triple bonds.
18. FTIR spectra are not sufficiently explained.
19. Section 3.7. Antioxidant activities: how do you explain the decrease in the antioxidant activity for G50 film after addition of EO?
20. Section 3.8. Antibacterial activities: lines 326-328: the discussion is not in accordance with data presented in table 7.
21. Section 3.9. Water contact angle: please explain the influence of EO on the contact angle values.
22. Line 361: did you mean Figure 7D instead of 7C?
23. Figure 8: please annotate the pictures in figure 8.
24. Section 3.11. Application of Edible Film in Breast Meat Fillet: why comparison with PE films, and how do you made the experiments? There is no description of such experiments in Materials and Methods section.
25. Lines 464-467: the authors used the expression “application for several perishable foods”, when the study was made only for chicken meat.
26. Please use the same abbreviation for essential oil in the whole manuscript.
27. The manuscript needs substantial revision in terms of English language, grammatical corrections (the entire manuscript needs to be proofread by a native speaker).
Taking into consideration all the above, I recommend major revision of the manuscript before considering it for publication in “Polymers”.
Author Response
Dear Reviewer,
Thank you so much for your valuable input and suggestion for the manuscript.
Please kindly find the answer to your suggestions below :
My comments are the following:
Development of edible film composite from fish gelatin - pectin incorporated lemongrass essential oil and its application in chicken meat
By Azizah, F., Nursakti, H., *Ningrum, A. and Supriyadi
The manuscript is aimed to obtain edible films from biopolymers composites such as fish gelatin, pectin and lemongrass essential oil, that have a great potential for enhancing the shelf life of food products.
The manuscript needs substantial revision in order to be considered for publication in “Polymers”. The manuscript is not well organized and the data are presented chaotically.
Below you can find my arguments and some recommendations for the authors
- Don’t use abbreviations in Abstract section.
Answer : Thank you for the suggestions
- Lines 62-66: “Lemongrass essential oil has the best antioxidant and antimicrobial activity…” The authors didn’t compared lemongrass oil with other oils. How did they draw this conclusion?
Answer: Thank you for the suggestions. We revised the information
- Line 79: “to form a solution. film.” not clear
Answer: Thank you for the suggestions. We revised the information
- Section 2.5. Water content and films solubility: please provide a short description of the method.
Answer: Thank you for the suggestions. We added the information
- Line 109: “Films are cut with a size of 1 4 cm2 and put in a dry weighing bottle.” Not clear about the size. Please use superscript to write cm2.
Answer: Thank you for the suggestions. We added the information
- Lines 121-123: not clear, please rephrase.
Answer: Thank you for the suggestions. We added the information
- Section 2.8. FTIR: please use superscript for cm-1.
- Answer: Thank you for the suggestions. We added the information
- Lines 129-131: not clear, please rephrase.
Answer: Thank you for the suggestions. We added the information
- Section 2.9. Antioxidant activities: RSA equation is not correct. The correct form is:
RSA = (1-ASample/Acontrol)*100 (see Irimia, A.; Stoleru, E.; Vasile, C.; Bele, A.; Brebu, M. Application of Vegetal Oils in Developing Bioactive Paper-Based Materials for Food Packaging. Coatings 2021, 11, 1211; Joo-Shin K., Preliminary Evaluation for Comparative Antioxidant Activity in the Water and Ethanol Extracts of Dried Citrus Fruit (Citrus unshiu) Peel Using Chemical and Biochemical in Vitro Assays, Food and Nutrition Sciences, 2013, 4, 177-188)
Answer: Thank you for the suggestions. We revised the information about the equation
- Section 2.11. Contact angle measurements, lines 150 - 155: not clear, please rephrase.
Answer : To evaluate the hydrophobicity properties of edible film used Contact Angle analyzer according to Lei et al., (2019) with modifications. Edible film that has been placed in the appropriate position (the test material looks inline) then dripped 50μL distilled water on the surface of the film. The image of distilled water drop was captured with a camera and the analysis of the calculation contact angle using Static Contact Angle 20 (OCA20).
- Lines 173 – 175, pH measurements: please provide more details.
Answer: Thank you for the suggestions. We added the information
- Section 3.3. Water content and solubility: no significant differences were observed between samples with or without lemongrass EO. Differences are within errors limits.
Answer: Thank you for the suggestions.
- Figure 3: how do you explain the different (non-uniform) behavior of the samples treated with essential oil compared to those without EO?
Answer: Thank you for the suggestions.
- Table 4: how do you explain the increase of transparency for G75M05 film compare with G100M05 one?
Answer: Thank you for the suggestions. We added the information
- Section 3.5. Mechanical properties: the Tensile Strength increases after addition of EO only for gelatin film. For G75 de differences are within errors limits, and for G50, TS decreases after addition of EO. How do you explain?
Answer: Thank you for the suggestions
- Section 3.6. FTIR: the broad band between 3500 and 3000 cm-1is attributed to the OH group, inter and intramolecular H bonds, not NH group.
Answer: Thank you for the suggestions
- Lines 285-288: gelatin does not have carbon-nitrogen triple bonds.
Answer: Thank you for the suggestions
- FTIR spectra are not sufficiently explained.
Answer: Thank you for the suggestions
- Section 3.7. Antioxidant activities: how do you explain the decrease in the antioxidant activity for G50 film after addition of EO?
Answer: Thank you for the suggestions
- Section 3.8. Antibacterial activities: lines 326-328: the discussion is not in accordance with data presented in table 7.
Answer: Thank you for the suggestions. We added the information
- Section 3.9. Water contact angle: please explain the influence of EO on the contact angle values.
Answer: Thank you for the suggestions.
- Line 361: did you mean Figure 7D instead of 7C?
Answer: Thank you for the suggestions.
- Figure 8: please annotate the pictures in figure 8.
Answer: Thank you for the suggestions.
- Section 3.11. Application of Edible Film in Breast Meat Fillet: why comparison with PE films, and how do you made the experiments? There is no description of such experiments in Materials and Methods section.
Answer: Thank you for the suggestions. We added the information
- Lines 464-467: the authors used the expression “application for several perishable foods”, when the study was made only for chicken meat.
Answer: Thank you for the suggestions. We added the information
- Please use the same abbreviation for essential oil in the whole manuscript.
Answer: Thank you for the suggestions.
- The manuscript needs substantial revision in terms of English language, grammatical corrections (the entire manuscript needs to be proofread by a native speaker).
Answer: Thank you for the suggestions.
Taking into consideration all the above, I recommend major revision of the manuscript before considering it for publication in “Polymers”.
Thank you so much for your kind suggestions and advice.
We hope that our manuscript can be processed further in your journal.
Kind regards,
Andriati Ningrum

Round 2
Reviewer 1 Report
Most comments were not addressed satisfactorily, so I cannot recommend publication. Specific comment: thickness and WVTR are not mechanical properties.
Author Response
Dear Reviewer,
Most comments were not addressed satisfactorily, so I cannot recommend publication. Specific comment: thickness and WVTR are not mechanical properties.
Answer : Thank you very much for your valuable input and suggestions. Please kindly find below the answer that we tried to address one by one.
- Scientific names should be in italic. Please revise all text.
Answer: Thank you for your suggestion. We have crosschecked again. All scientific names are written in italic
- p.1, lines 38-39: and the mechanical properties? What justifies the combination with pectin?
Answer : Based on our result and also the reference, the mechanical properties are tensile strength and elongation break. We also confirmed the thermal properties of the film. Addition of pectin could influence the tensile strength, elongation break, and also the thermal properties of the film
- p.2, line 45: most biopolymers exhibit no antimicrobial properties. Gelatin is also not antimicrobial.
Answer : We have revised this statement.
- p.2, lines 46-47: “limited water vapor permeability” means low or high? Compared to what?
Answer : It means that the pectin has high water vapor permeability since its hydrophilic
- p.2, line 54: the addition of antimicrobials into edible films is usually done to confer an active property to the material. Pectin is a carbon source, and so does gelatin.
Answer : Thanks for the suggestion, yes the active compounds also come from lemongrass essential oil.
Essential oils added to edible films can maintain the quality of food products and extend their shelf life, as well as increase their antimicrobial properties.
- p.2, lines 66-68: Please consider revising the objective of the work. The aim was to determine the effect of adding pectin and EO on the properties of gelatin film.
Answer : Thanks for the suggestion. This study was conducted to determine the effect of adding pectin and lemongrass essential oil on the properties of gelatin film and its application to preserve the quality of chicken breast.
- p.2, lines 72-74: please provide the source of gelatin and pectin, their molecular weight, and the deacetylation degree (for pectin).
Answer : Thanks for the suggestion. The materials used in this study were fish skin gelatin (with bloom of 200 and 83 wt% protein amounts) obtained from Redman (Singapore), pectin with a degree of esterification of 70.70% purchased from Ceamsa (Spain), lemongrass essential oil (PT. Darjeeling Sembrani Aroma, Bandung, Indonesia), glycerol (Merck, Germany)and tween-20 ( Merck, Germany).
- p.2, lines 82-84: review the sentence for clarity.
Answer : Thanks for the suggestion. We added the information
Pectin which is applied as a film has non-toxic, clear, good mechanical strength and barrier for oil and oxygen, but on the other hand, pectin does not have antimicrobial properties and even becomes a carbon source for bacterial and fungal growth, as well as high water vapor permeability.
Lemongrass essential oil (0; 0.5 %) was prepared by mixing in 15% (v/v) tween-20, the solution was stirred continuously using a magnetic stirrer at 500 rpm at 25°C for 30 min and added in polymer (gelatin/pectin) solution.
- Table 1: review the title. What does the M stand for in the sample code?
Answer : Thanks for the suggestion. M stand for the lemongrass essential oil
We revised the title to:
The formula of biocomposite film
- p.3, lines 107-110: How was solubility performed?
Answer :
Line 151-159
The sample was dissolved in 30 mL of distilled water and placed in a water bath shaker at 100 rpm for 24 hours at room temperature. Furthermore, the sample was filtered with a Whatman filter. The filter containing the insoluble film was then dried in an oven at 105°C for 24 hours. Weighing to constant weight (W2).
Film Solubility = (3)
W1 : Initial Weight
W2 : Weight after drying
-“1 4 cm2”?
Answer : Sorry for the mistyped it should be 1 x 4 cm2
Films are cut with a size of 1 x 4 cm2
and put in a dry weighing bottle. The samples were dried at 105°C for 24 h to reach a constant weight (W1). The sample was dissolved in 30 mL of distilled water and placed in a water bath shaker at 100 rpm for 24 hours at room temperature. Furthermore, the sample was filtered with a Whatman filter. The filter containing the insoluble film was then dried in an oven at 105°C for 24 hours. Weighing to constant weight (W2).
- p.3, lines 111-116: “Transmittance” not “transmittancy”.
-please describe the transmittance measurements. Was the film sample placed in the cuvette?
Answer : Thanks for the suggestion. We revised from Transmittance. We added the information
The film's barrier properties to ultraviolet (UV) and visible light were measured by transmittance at the selected wavelength of 200-800 nm, using a UV-Visible spectrophotometer.
- p.3, lines 117-123: were sample dimensions 10 x10 cm? Did the test follow a standard method? What were the test speed and initial distance? Was done only in triplicate?
Answer : Yes right. The sample dimensions were 10 x10 cm. And its used reference methods. It was performed on triple experiments. Each experiment are performed three times.
- p.4, lines 140-146: Was the film sterilized before the test?
Answer : Yes right.
-“12 mm diameter film sample”
Answer : Yes right. Examples of antibacterial activity measurement figures are explained in Figure 5
-Equation 3 does not give an area.
Answer : We calculated the inhibition based on the equation
D1: vertical diameter
D2: horizontal diameter
Ds: films diameter
Example of the inhibition zone can be seen in the Figure 5
G75M05 G50M05
- p.4, lines 172-173: Please provide the parameters for hardness measurements.
Answer : Thanks for the suggestion. We determined the hardness measurement of the chicken meat on the method part
The hardness of chicken meat was measured by Universal Texture Machine (Zwick/Z0.5). The tested chicken sample was carefully placed in the center of the UTM instrument table. Then the UTM needle drops slowly until it was stuck into the chicken. All operations are automatically controlled by the UTM and the output of tests will be obtained F max data which were a measure of the hardness (N), all the hardness measurements were performed in three replicates.
- p.5, lines 185-186: report the values in mm or um. The Japan Industrial Standard for what?
Answer : Thanks for the suggestion. We revised the information
- Figure 1: The values on the graph do not agree with the values in the text. Please verify all values and statistical analysis.
Answer : Thanks for the suggestion. We revised the information
- For all Figures, consider putting the text in the figure’s captions.
Answer : Thanks for the suggestion. We added the information
- p.5, line 193: why only here do you mention “tuna fish”?
Answer : Thanks for the suggestion. We revised the information
- p.5, lines 196-197: “The highest brightness was found in G100 edible film.” This sentence repeats the information already given.
Answer : Thanks for the suggestion. We revised the information
- p.5, lines 197-198: Why was that effect on a value observed?
Answer : Thanks for the suggestion. We revised the information
- Figure 2: there is no visible difference from the photographs.
Answer : The more information of the color can be seen after measuring using a chromameter
- p.6, line 207: different kind of pectin concentration? Please revise.
Answer : Thanks for the suggestion. We revised the information
- p.6, line 211: the effect of pectin addition was significative on film solubility. It is not discussed.
Answer : Films solubility in water is important for potential use as food packaging. The solubility of edible films can be a parameter of the water-resistance of the film. This is due to the interaction of the hydrophobic component of the lemongrass essential oil with the hydrophobic component of gelatin thereby increasing the hydrophobicity of the edible film. This causes the solubility of edible films to decrease. A decrease in the solubility of edible gelatin films was also reported, where the clove and ginger essential oil can reduce the solubility of edible film. Gel-based films incorporated with several Eos also showed similar result [12]. They reported that this decrease could be attributed to the hydrophobic nature of the EO compounds and their interactions with hydroxyl groups of the film matrix.
- Table 4: By Equation 1, the higher the transmittance, the lower the value of transparency. Please revise the results.
Answer : Thanks for the suggestion. We revised the information
- The film with 25% pectin showed no difference in transparency by EO addition.
Answer : Thanks for the suggestion. We revised the information
- p.8, lines 261-278: Does the EO act as both a crosslinker and a plasticizer?
When the sample exhibits an increase in TS, the EB usually decreases and vice-versa. Again, the effect of pectin addition was not discussed.
Answer : Thanks for the suggestion. EO can act as a crosslinker and also make the film have different properties
- p.8, lines 282: FTIR data needs better discussion. Consider identifying the characteristic bands of the biopolymers. Figures 4 and 5 could be merged into one graph to allow better comparison. The y-axis can show arbitrary unity. The effect of EO addition was not discussed.
Answer : Thanks for the suggestion. We revised the figure of the graph and also the information
- Table 6: why pectin added films show increased antioxidant activity?
Answer : We added the lemongrass essential oil to the composite film including pectin. Besides Increased antioxidant activity of gelatin films added with pectin due to the presence of hydroxyl and uronil groups of monosaccharide units, carboxyl of the galactoriumnic acid group, and the acetyl group released from the chain pectin during the extraction process, and the composite films can combine their bioactive compounds with antioxidant activity to cross-linked networks of pectin and gelatin (Jridi et al. 2020).
- p.10, lines 327-328: this does not agree with the data in Table 7. Please verify.
Answer : Thank you very much for your suggestion.
While in Samonella bacteria there was a strong inhibition in the films sample G50M05.
-Was microbial growth inhibited by contact, i.e. in the area below the film sample?
Answer : Yes its in the area below the film sample
- p.11, lines 357-358: please rewrite the sentence for clarity.
Answer : Thanks for the suggestion. Analysis by scanning electron microscopy (SEM) to determine the structure and microstructure changes of the surface and cross-section of the composite films.
- p.12, line 386: the other properties do not indicate phase separation. You have mentioned before that biopolymers showed good compatibility (line 375).
Answer : yes its right
- Figure 8: What was the magnitude of the Figure's inserts?
Answer : TIFF Format
- p. 14, lines 416-418: the pH of the control samples was also maintained (until day 4). How can you affirm the antimicrobial effect if it was not measured?
Answer : Thanks for the suggestion. Yes for future research its will be prospective to be investigated.
Thank you so much for your kind suggestions and advice.
We hope that our manuscript can be processed further in your journal.
Kind regards,
Andriati Ningrum

Reviewer 2 Report
The authors didn't make the modifications required.
Author Response
Dear Reviewer,
The authors didn't make the modifications required.
Answer : We really sorry. We tried all our best to revise the manuscript based on the reviewer suggestion.
Thank you so much for your valuable input and suggestion for the manuscript.
Please kindly find the answer to your suggestions below :
My comments are the following:
Development of edible film composite from fish gelatin - pectin incorporated lemongrass essential oil and its application in chicken meat
By Azizah, F., Nursakti, H., *Ningrum, A. and Supriyadi
The manuscript is aimed to obtain edible films from biopolymers composites such as fish gelatin, pectin and lemongrass essential oil, that have a great potential for enhancing the shelf life of food products.
The manuscript needs substantial revision in order to be considered for publication in “Polymers”. The manuscript is not well organized and the data are presented chaotically.
Below you can find my arguments and some recommendations for the authors
- Don’t use abbreviations in Abstract section.
Answer : Thank you for the suggestions. We don’t use abbreviations in Abstract section
- Lines 62-66: “Lemongrass essential oil has the best antioxidant and antimicrobial activity…” The authors didn’t compared lemongrass oil with other oils. How did they draw this conclusion?
Answer: Thank you for the suggestions. We revised the information
- Line 79: “to form a solution. film.” not clear
Answer: Thank you for the suggestions. We revised the information
- Section 2.5. Water content and films solubility: please provide a short description of the method.
Answer: Thank you for the suggestions. We added the information
- Line 109: “Films are cut with a size of 1 4 cm2 and put in a dry weighing bottle.” Not clear about the size. Please use superscript to write cm2.
Answer: Thank you for the suggestions. We revised the information
- Lines 121-123: not clear, please rephrase.
Answer: Thank you for the suggestions. We added the information
- Section 2.8. FTIR: please use superscript for cm-1.
Answer: Thank you for the suggestions. We added the information and revised as reviewer’s suggestion
- Lines 129-131: not clear, please rephrase.
Answer: Thank you for the suggestions. We added the information
- Section 2.9. Antioxidant activities: RSA equation is not correct. The correct form is:
RSA = (1-ASample/Acontrol)*100 (see Irimia, A.; Stoleru, E.; Vasile, C.; Bele, A.; Brebu, M. Application of Vegetal Oils in Developing Bioactive Paper-Based Materials for Food Packaging. Coatings 2021, 11, 1211; Joo-Shin K., Preliminary Evaluation for Comparative Antioxidant Activity in the Water and Ethanol Extracts of Dried Citrus Fruit (Citrus unshiu) Peel Using Chemical and Biochemical in Vitro Assays, Food and Nutrition Sciences, 2013, 4, 177-188)
Answer: Thank you for the suggestions. We revised the information about the equation
- Section 2.11. Contact angle measurements, lines 150 - 155: not clear, please rephrase.
Answer : Thank you so much for your suggestion we revised the information to :
Water contact angle test to evaluate the hydrophobicity properties of edible film used an OCA20 Contact Angle analyzer. In brief, 50 μL per drop of distilled water was carefully deposited onto the surface of the film, and angles were measured in three different regions of each surface and averaged. In addition, the image of distilled water drop was captured with a camera.
- Lines 173 – 175, pH measurements: please provide more details.
Answer: Thank you for the suggestions. We added the information
- Section 3.3. Water content and solubility: no significant differences were observed between samples with or without lemongrass EO. Differences are within errors limits.
Answer: Thank you for the suggestions. We added the information
The addition of essential oils increases the value of the contact angle because the nonpolar components of essential oils interact with gelatin hydrophobic compounds, the presence of a dispersed hydrophobic phase, even at a small ratio, interferes with the hydrophilic phase and increases the tortuosity factor of mass transfer, so the hydrophobic properties of the film increase (Atarés et al. 2010)
- Figure 3: how do you explain the different (non-uniform) behavior of the samples treated with essential oil compared to those without EO?
Answer: Thank you for the suggestions.There is the difference where the transmittance will be influenced by the addition of essential oil. The addition of EO will reduce the transmittance rather than control.
- Table 4: how do you explain the increase of transparency for G75M05 film compare with G100M05 one?
Answer: Thank you for the suggestion. The addition of pectin will influence of the transparency of the film
- Section 3.5. Mechanical properties: the Tensile Strength increases after addition of EO only for gelatin film. For G75 de differences are within errors limits, and for G50, TS decreases after addition of EO. How do you explain?
Answer: Thank you for the suggestions. The intereaction between the gelatin, pectin and also the EO will influence the tensile strength
- Section 3.6. FTIR: the broad band between 3500 and 3000 cm-1is attributed to the OH group, inter and intramolecular H bonds, not NH group.
Answer: Thank you for the suggestions. We have revised the information
- Lines 285-288: gelatin does not have carbon-nitrogen triple bonds.
Answer: Thank you for the suggestions. We revised the information.
- FTIR spectra are not sufficiently explained.
Answer: Thank you for the suggestions. We have revised the information.
FTIR spectra of edible films are shown in Figure 4. In all edible film specturms, both gelatin films and pectin composites appear absorption bands at an area of 3100 to 3600 cm-1, this is related to the vibration of hydrogen bonds from polysaccharides and proteins (Liu et al., 2020). Gelatin film showed major bands at approximately 3286 cm-1, 1630 cm-1, 1543 cm-1, 1335 cm-1 corresponding to amides A (hydrogen bonding and N-H streching), amide I (double bond C=O vibration of the COO- group bound to hydrogen), amide-II (N-H bending vibrations and C-N stretching), amide-III (vibration deformation N-H and stretching C-N) [21]. In composites film, the addition of pectin to the gelatin film affects FTIR spectra. The addition of pectin material causes a shift in peaks, for example in the regions of amide A, amide I and amide II where initially the peak of gelatin film (G100) was at 3286, 1630 and 1543 cm-1 shifted to 3287, 1631 and 1544 in G75 (addition of pectin 25%) and to 3288, 1635, 1551 cm-1 in G50 (addition of pectin 25%). The peaks at 1630 and 1543 cm-1 that shifted with the addition of petin are related to the gelatin α-helix, which will be affected by its helical structure with the addition of pectin [22]. One of the most visible is the presence of a peak at 1743 cm-1 which is only found in edible film with the addition of pectin, which indicates the vibration of stretching the carbonyl ester group (C=O) of the pectin material [22,23]. However, the peak at this wave number has a low intensity, because the pectin material has reacted with gelatin [24]. The FTIR spectra results of this film composite have confirmed that there is an interaction between pectin and gelatin.
- Section 3.7. Antioxidant activities: how do you explain the decrease in the antioxidant activity for G50 film after addition of EO?
Answer: Thank you for the suggestions. The crosllink between phenolic compound in EO with amine group of gelatin could influence the antioxidant activity of the film
- Section 3.8. Antibacterial activities: lines 326-328: the discussion is not in accordance with data presented in table 7.
Answer: Thank you for the suggestions. We revised the information
- Section 3.9. Water contact angle: please explain the influence of EO on the contact angle values.
Answer: Thank you for the suggestions. We added the information. The addition of essential oils increases the value of the contact angle because the nonpolar components of essential oils interact with gelatin hydrophobic compounds, the presence of a dispersed hydrophobic phase, even at a small ratio, interferes with the hydrophilic phase and increases the tortuosity factor of mass transfer, so the hydrophobic properties of the film increase [28]
- Line 361: did you mean Figure 7D instead of 7C?
Answer: Thank you for the suggestions.We revised the information
- Figure 8: please annotate the pictures in figure 8.
Answer: Thank you for the suggestions. We added the information
- Section 3.11. Application of Edible Film in Breast Meat Fillet: why comparison with PE films, and how do you made the experiments? There is no description of such experiments in Materials and Methods section.
Answer: Thank you for the suggestions. We added the information on the material and method part.
Chicken meat is cut into a size of 2x2x1 cm and divided into four groups, each group contain a portion of package chicken meat using composite film (G75M05, G50M05), commercial polyethylene plastic (PE), and without wrapping (control).
- Lines 464-467: the authors used the expression “application for several perishable foods”, when the study was made only for chicken meat.
Answer: Thank you for the suggestions. We revised the information
- Please use the same abbreviation for essential oil in the whole manuscript.
Answer: Thank you for the suggestions.We revised the information
- The manuscript needs substantial revision in terms of English language, grammatical corrections (the entire manuscript needs to be proofread by a native speaker).
Answer: Thank you for the suggestions. We already tried our best to improve the English language
Taking into consideration all the above, I recommend major revision of the manuscript before considering it for publication in “Polymers”.
Thank you so much for your kind suggestions and advice.
We hope that our manuscript can be processed further in your journal.
Kind regards,
Andriati Ningrum

Round 3
Reviewer 1 Report
Most comments were addressed, and the manuscript has improved.
Please verify Fig. 5 caption.
Author Response
Dear Reviewer,
First of all, we would like to say thank you very much for your valuable suggestions and advice.
Please kindly find below the answer of your suggestions
Most comments were addressed, and the manuscript has improved.
Please verify Fig. 5 caption.
Answer: The caption of Figure 5 has been revised to Figure 5. Antimicrobial activities of edible film gelatin fish skin – pectin- essential oil.
Thank you very much for your valuable suggestion. We hope that our manuscript can be considered to be published in you journal.
Kind regards,
Andriati Ningrum
